# High-Strength Heat-Elongated Thermoplastic Polyurethane Elastomer Consisting of a Stacked Domain Structure

**DOI:** 10.3390/polym14071470

**Published:** 2022-04-04

**Authors:** Mutsumi Takano, Koudai Takamatsu, Hiromu Saito

**Affiliations:** Department of Organic and Polymer Materials Chemistry, Tokyo University of Agriculture and Technology, Koganei-shi, Tokyo 184-8583, Japan; s203108w@st.go.tuat.ac.jp (M.T.); s197310z@st.go.tuat.ac.jp (K.T.)

**Keywords:** polyurethane, heat elongation, tensile property, stacked domain structure, small-angle X-ray scattering

## Abstract

We found that a high-strength elastomer was obtained by the heat elongation of a thermoplastic polyurethane (TPU) film consisting of a high content of crystalline hard segments (HS). The stress upturn continuously increased with the elongation ratio without a decrease in the strain recovery by heat elongation, i.e., the stress at break of a quenched TPU film was increased from 55 to 136 MPa by heat elongation at an elongation ratio of 300%. The results of small-angle X-ray scattering, DSC, and AFM observations revealed that: (1) anisotropically shaped HS domains were stacked at a nanometer scale and the longer direction of the HS domains was arranged perpendicular to the elongated direction due to the heat elongation, (2) the densification of the HS domains increased with increases in the elongation ratio without a significant increase in the crystallinity, and (3) the stacked domain structure remained during the stretching at 23 °C. Thus, the strengthening of the elongated TPU might be attributed to the densification of the HS domains in the stacked structure, which prevents the fracture of the HS domains during the stretching.

## 1. Introduction

Thermoplastic polyurethane (TPU) is one of the most-used elastomers, with versatile thermoplastic and elastomeric properties [1]. TPU has superior mechanical properties such as high elongation, toughness, and strain recovery [2,3,4,5]. The characteristic mechanical behavior of TPU is attributed to the sequenced crystalline hard segment (HS) and amorphous soft segment (SS). The HS and SS are self-assembled at a nanometer scale due to microphase separation caused by the thermodynamic incompatibility between them. The microphase-separated structure depends on the HS content [6,7,8,9,10] and chemical structure [11,12,13,14,15,16], and can be changed by thermal annealing [17,18,19,20,21,22]. Since the HS domain acts as a physical cross-link point, the tensile property is correlated with the microphase-separated structure. The Young’s modulus and dynamic strange modulus increase because of the densification of the HS domain due to the ordering and packing of the HS chains [23,24,25,26,27], and the stress at break increases according to the orientation of the HS chains parallel to the stretching direction [28,29,30].

The structure change of TPU during uniaxial stretching is significant for the tensile property. A wide-angle X-ray diffraction (WAXD) study indicated that the stress upturn observed in the stress–strain behavior at high strain could be attributed to the strain-induced crystallization [31,32,33,34,35]. The four-point pattern observed by small angle X-ray scattering (SAXS) measurement suggests that the HS domain is tilted along the stretching direction (SD) during uniaxial stretching [32,35,36,37,38,39], and a streak SAXS pattern observed in the perpendicular direction to the SD indicates that the HS domains are broken and fibrillated in the stretching direction [31,32,35,36,40]. The fibrillation causes a gradual increase in stress at the stress upturn [35]. This is required to obtain a high-strength TPU elastomer for widespread applications such as industrial belt and medical tube, in which a light weight and thin structure are desired. However, the stress at break of the TPU film is usually below 50 MPa [11,14,23,26,41], and the literature on high-strength TPU that can achieve stress at break above 100 MPa is limited [29].

Magnesium alloys consisting of mutually stacked hard and soft layers were found to exhibit strengthening by uniaxial stretching perpendicular to the longer direction of the stacked layers [42,43,44]; this mutually stacked layer structure is called a mille-feuille structure [45,46,47]. Strengthening was also found to occur in the mille-feuille-structured, high-density polyethylene (HDPE) film consisting of stacked hard crystalline lamella layers and soft amorphous ones due to the suppression of the yielding behavior preventing the fracture of lamellae and void formation [48].

We found strengthening of the TPU film consisting of the stacked HS domain structure to be caused by heat elongation during our investigation of the tensile properties, i.e., the stress at break increased from 55 to 136 MPa without a sacrifice in the strain recovery following a heat elongation of 300%, although the stress at break of the TPU film is usually below 50 MPa. In this paper, we report the strengthening of the TPU film caused by heat elongation, depending on the elongation ratio. The TPU used in this study consists of crystalline 4,4′-diphenylmethane diisocyanate (MDI) and 1,4-butandiol (BD) as HS with a weight fraction of 75% and amorphous poly(tetramethylene ether) glycol (PTMG) as SS. The tensile properties of the heat-elongated TPU film are discussed in terms of the domain structure estimated by SAXS, WAXD, differential scanning calorimetry (DSC) and atomic force microscopy (AFM).

## 2. Experimental Section

### 2.1. Preparation of the Specimen

The thermoplastic polyurethane (TPU) pellets used in this study were MIRACTRAN E398, supplied by Nippon Miractran Company Limited. This TPU consisted of amorphous soft segment (SS) blocks made of polyester glycol, crystalline hard segment (HS) blocks made of MDI and the chain extender 1,4-BD with sequence length distributions. The weight fraction of the HS in the TPU, which is the sum of the weight fraction of MDI and BD characterized by NMR, was 75%.

The pellets were dried under vacuum (10^−4^ mmHg) at 100 °C for a day. The pellets were melted in a small hot-press machine (11FD, Imoto Machinery Co., Ltd., Kyoto, Japan) at 250 °C and pressed at 40 MPa for 5 min under vacuum to obtain the film specimen. Then, the melt specimen was quenched in an ice-water bath. The film specimen was called a quenched TPU film. The elongated TPU film was obtained by heat elongation of the quenched TPU film at 150 °C at a constant rate of 50 mm/min, up to various elongation ratios *λ* and up to 300% in a heat-stretching apparatus (Taiatsu Techno Corporation, Tokyo, Japan). The elongated TPU film was gradually cooled by air after the heat elongation. The results of *λ* = 100%, 200%, and 300% are shown in this paper.

### 2.2. Tensile Tests

For the tensile test, the dumbbell-shaped film specimen was prepared by die-cutting for a dumbbell according to JISK6251-7, in which the length and width of the narrow section were 12.0 mm and 2.0 mm, respectively, the overall length was 35.0 mm and the overall width was 6.0 mm. The stress–strain curve of the film specimen was obtained by stretching at 23 °C using an Instron-type tensile testing machine (Strograph VES05D, Toyo Seiki Co., Ltd., Tokyo, Japan) at a crosshead speed of 100 mm/min. The deformed length during the stretching was obtained by measuring the distance between the marked positions using a measure. The nominal stress σ and the nominal strain *ε* were obtained by:(1)$$\sigma =\frac{F}{A_{0}}$$
(2)$$\varepsilon \left(\%\right)=\frac{L-L_{0}}{L_{0}}\times 100$$
where *F* is the load, *A*_0_ is the cross-sectional area of the specimen before deformation, *L* is the deformed length, and *L*_0_ is the initial length. After stretching, the specimen was released from the clamps and the residual strain was measured at certain intervals.

### 2.3. SAXS and WAXD Measurement

SAXS and WAXD measurements were performed by NANO-Viewer system (Rigaku Co., Ltd., Tokyo, Japan). Cu-Ka radiation with a wavelength λ_0_ of 0.154 nm was generated at 46 kV and 60 mA and collimated by a confocal max-flux mirror system. Measurements were performed at room temperature and the exposure times were 1 h for SAXS and 5 min for WAXD measurements. The samples were stretched in steps at fixed strain using a miniature stretching machine (TensilIR, S.T. Japan Inc., Tokyo, Japan) for measurements during stretching. The X-ray was radiated at the midpoint of the stretched area by moving the two crossheads. An imaging plate (IP) (BAS-SR 127, Fujifilm Co., Ltd., Tokyo, Japan) was used as a two-dimensional detector and the IP-reading device (R-AXIS Ds3, Rigaku Co., Ltd., Tokyo, Japan) was used to transform the obtained scattering images into the text data. The scattering intensity was corrected with respect to the exposure time, the thickness of the specimen, and the transmittance.

### 2.4. DSC Measurement

DSC measurements were performed by heating the sample in the pan at a heating rate of 10 °C/min in a 1st cycle under a nitrogen atmosphere using a calorimeter (DSC-Q200, TA Instruments, New Castle, DE, USA). The weight of the sample used for the measurement was about 4 mg.

### 2.5. AFM Observation

The AFM observation for the phase image was performed using a Shimadzu SPM-9700HT (Shimadzu Corporation, Kyoto, Japan) using an OMCL-AC160TS-C3 silicone cantilever (Olympus, Tokyo, Japan) with a radius of 7 nm, a spring constant of 26 N/m, and a resonance frequency of 300 kHz. The phase image was obtained by the phase lag between the output signal and the cantilever oscillation [49]. The specimen preparation method was the same as that used for other measurements.

## 3. Results and Discussion

### 3.1. Tensile Properties of the Elongated TPU

Figure 1 shows the stress–strain curves of the heat elongated TPU film obtained by elongation at 150 °C at various elongation ratios *λ*. The stress–strain curve of the quenched TPU film before elongation (*λ* = 0%) is also shown in Figure 1 for comparison. In the quenched TPU, the stress increased steeply at strain *ε* below 10% (*ε* < 10%) and then increased slightly at 20% < ε < 100%. After the slight increase in the stress, the stress upturn occurred at around *ε* = 100%, and the stress gradually increased from 15 to 55 MPa with the increase in *ε* up to the breaking point at around *ε* = 320%. The interesting result here is that the stress drastically increased with the increase in *λ* due to the heat elongation of the quenched TPU. The stress upturn started to occur at lower *ε* with increasing *λ* in the heat-elongated TUP. The stress increase at the stress upturn continuously increased with *λ*, although the increase in the initial increase with *λ* under low strain (*ε* < 10%) was slight. Due to the drastic change in the stress increase with *λ* at the stress upturn, the stress at break of the quenched TPU film increased from 55 to 136 MPa following heat elongation at *λ* = 300%, i.e., the enhancement of the stress by the heat elongation was about 2.5 times.

Despite the strengthening, the strain recovery of the heat-elongated TPU was excellent. Retraction occurred immediately upon releasing the heat-elongated TPU after stretching. The stretched strain of *ε* = 100% was immediately recovered to *ε* = 5% after releasing the quenched TPU and the heat-elongated TPU (*λ* = 300%), and the residual strain was almost 0% at 20 min in the quenched and elongated TPU. The results indicate that the strengthening occurred without sacrificing strain recovery in the heat-elongated TPU film. Thus, a high-strength TPU elastomer could be obtained by the heat elongation. Since the tensile property continuously changed with *λ*, the TPU elastomers with various tensile properties could be obtained, i.e., a high-strength elastomer could be obtained by a heat elongation at a large *λ,* while a soft TPU elastomer could be obtained by a heat elongation at a small *λ.*

### 3.2. Structure Change during the Heat Elongation

Figure 2 shows the two-dimensional (2D) SAXS patterns of the elongated TPU films at various elongation ratios *λ*. A weak isotropic ring pattern was observed in the quenched TPU film (*λ* = 0%), indicating that the microphase-separated structure at a nanometer scale consisting of a hard segment (HS) and soft segment (SS) is regularly arranged at a periodic distance (Figure 2a). Following elongation at *λ* = 100%, the isotropic ring pattern changed to a curved-layer one (Figure 2b). According to the Metropolis Monte Carlo simulation for an SAXS pattern using a small-particle aggregation model, the curved-layer pattern is attributed to the stack of an anisotropically shaped domain structure [50]. Thus, the appearance of the curved-layer pattern indicates that the macroscopically arranged stacked domain structure consisting of anisotropically shaped domains is formed by the heat elongation, and the longer direction of the HS domains is arranged perpendicular to the elongated direction (ED). The curved-layer patterns were maintained despite the further elongation and the scattering intensity increased with the increasing *λ* (Figure 2c,d). This stacked domain structure might be similar to the mille-feuille structure consisting of alternatively stacked crystalline layer and the amorphous one suggested by the layer-shaped SAXS pattern observed in the heat-elongated HDPE film, which exhibited strengthening [48].

Figure 3 shows the one-dimensional (1D) SAXS profiles of the heat-elongated TPU films derived from the 2D SAXS patterns shown in Figure 2. The SAXS profile *I*(*q*) shown in Figure 3a was obtained at the azimuthal angle of −10–10° of the SAXS patterns in Figure 2, where 0° is the meridional direction, and the *q* is the scattering vector defined as *q* = 4π/*λ*_0_ sin *θ,* where *λ*_0_ is the wavelength of X-ray and *θ* is the scattering angle. A single peak was seen at the *q*_max_ in the *I*(*q*) in the quenched TPU film (*λ* = 0%), indicating that HS domains and SS ones are periodically stacked, i.e., the most probable next-neighbor distance of the HS domains in the elongated direction obtained by 2π/*q*_max_ was 8.8 nm. The peak position shifted slightly to a lower *q* with the increasing elongation ratio *λ* at *λ* > 100%, in which the layer pattern was observed, indicating that the change in the next-neighbor distance of the HS domains in the stacked structure was slight with *λ*, i.e., the next-neighbor distances obtained by 2π/*q*_max_ at *λ* = 100% and 300% were 11.3 nm and 11.9 nm, respectively. The peak intensity in the *I*(*q*) became stronger with *λ* (Figure 3a). The increase in peak intensity might be attributed to the increase in densification in the HS domains and crystallinity. The peak became sharper through association with the increased peak intensity, indicating that the periodicity of the HS domains in the stacked structure increases with *λ*. The peak profile for the azimuthal angle *ϕ* dependence of the scattering intensity *I*(*ϕ*) became sharper, and the peak intensity became stronger with the increasing *λ* (Figure 3b), indicating that the orientation degree of the HS domains increases with increasing *λ* according to the macroscopically arranged stacked domain structure.

The DSC curves at the endothermic peak, which are needed to melt the heat-elongated TPU films, are shown in Figure 4. The heat of fusion Δ*H* obtained by the peak area is shown in Figure 4. The change in Δ*H* was slight with an elongation ratio of *λ*, suggesting that the change in crystallinity was slight with *λ*, although the SAXS intensity increased, as shown in Figure 3a. Thus, the increase in the SAXS peak intensity shown in Figure 3a is not attributed to the increase in the crystallinity, but to the densification of the HS domains in the stacked structure.

Figure 5 shows the AFM phase image of the quenched TPU film (*λ* = 0%) and the heat-elongated one (*λ* = 300%). Here, the color depth was normalized by the phase difference, and the phase difference is ascribed to the hardness difference between the HS and SS phase. The hard region is indicated by red and green, while the soft region is indicated by blue. Connected HS domains were dispersed in the SS matrix in the quenched TPU (Figure 5a). In the elongated TPU, a stacked domain structure consisting of anisotropically shaped HS domains with a next-neighbor distance of tens of nanometers were seen, and a longer direction was arranged for anisotropically shaped HS domains perpendicular to the elongated direction (ED) (Figure 5b). These results support the macroscopically arranged stacked domain structure and arrangement of the anisotropically shaped HS domains perpendicular to the ED, as suggested by the SAXS patterns shown in Figure 2b–d. The next-neighbor distances in the elongated direction, estimated by AFM observation and SAXS measurements, had the same order, but did not exactly match due to the inclination of the crystallites [51]. The interesting result here is that the difference in hardness between the HS and SS phase drastically increased following the heat elongation of the quenched TPU. The increase in the hardness difference caused by heat elongation was attributed to the increase in the density difference due to the densification of the HS domains suggested by the increase in the SAXS intensity with *λ,* as shown in Figure 3a.

Figure 6 shows the schematic illustration of the TPU film during heat elongation at 150 °C. The regularly arranged microphase-separated structure of HS and SS domains in the quenched film (*λ* = 0%) are macroscopically arranged into a stacked domain structure consisting of the anisotropically shaped HS domains caused by heat elongation, and the longer direction of the HS domains are arranged perpendicular to the elongated direction (ED) (Figure 6a,b). The densification of the HS domain increases without a significant increase in the crystallinity during heat elongation (Figure 6c). The densification is ascribed to the ordering and packing of the HS chains, as suggested in references [23,24,25,26,27]. Since the densification in the HS layers increases with *λ*, the TPU strengthening shown in Figure 1 might be attributed to the existence of the macroscopically arranged stacked domain structure and the increase in the densification of the HS domains in the stacked structure.

### 3.3. Structure Change in the Elongated TPU during Stretching

Figure 7 shows the 2D SAXS patterns of the elongated TPU film (*λ* = 300%) during stretching at 23 °C with various strains of *ε*. A layer pattern was observed during stretching up to the breaking point, indicating that the stacked domain structure was not destroyed but remained up to the breaking point. The curved layer patten at *ε* = 0% (Figure 7a) changed to a straight-layer pattern with *ε* at the stress upturn (Figure 7b,c), suggesting that the HS domains in the stacked structure remained and the ordering of the arrangement of the domains increased during the stress upturn. The streak pattern caused by the fibrillation was not observed in the elongated TPU film after stretching, although fibrillation is considered to occur at the stress upturn [31,32,35,36,40].

The 1D SAXS profile in the stretching direction of the SAXS patterns of Figure 7a–c is shown in Figure 8 for various strains *ε*. Here, the SAXS profile *I*(*q*) was obtained at the azimuthal angle of −10–10° of the SAXS pattern, where 0° is the meridional direction. A single peak was seen, indicating that the HS and SS domains are periodically stacked. The peak was shifted to a lower *q* during the stretching, indicating that the most probable next-neighbor distance of the HS domain in the stacked structure increased during the stretching, i.e., the next-neighbor distance obtained by 2π/*q*_max_ increased from 12.4 nm at *ε* = 0% to 22.0 nm at *ε* = 100%. The peak became sharper, and the peak intensity increased with increasing *ε*. These results suggest that the stacked structure remains during the deformation at the stress upturn by preventing the fracture of the HS domains.

Figure 9 shows the 2D WAXD pattern of the elongated TPU film (*λ* = 300%) during stretching at 23 °C at various strains of *ε*. The spot pattern caused by strain-induced crystallization was not observed, but a broad, arc-shaped pattern was observed during the stretching up to breaking point. The arc-shaped pattern at *ε* = 0% became narrower with *ε* at the stress upturn, indicating that the crystalline HS chains are oriented with *ε* at the stress upturn by associating with the ordering of the HS domains in the stack structure. The increase in the sharpness of the 1D WAXD profile derived from Figure 9 was slight with *ε* at the stress upturn (Figure 10). Here, the 1D profile *I*(*θ*) shown in Figure 10 was obtained at the azimuthal angle of 80–100° of the WAXD patterns in Figure 9, where 0° is the meridional direction. These results indicate that strain-induced crystallization was not observed in the heat-elongated TPU at the stress upturn, although stress upturn is attributed to the strain-induced crystallization [31,32,33,34,35].

Thus, the stress increase at the stress upturn with *λ* in the elongated TPU film, as shown in Figure 1, is not attributed to the fibrillation and strain-induced crystallization, but is attributed to the increase in the densification of the HS domains tied in the stacked domain structure by the taut tie chain of the SS domains, which prevents the HS domains from fracturing during stretching due to the large energy storage that occurs during stretching to prevent the large deformation of the HS domains. This concept differs from familiar concepts regarding the strengthening at the stress upturn, which requires fibrillation, and the strain-induced crystallization of the HS domains, but is consistent with the strengthening mechanism suggested in the heat-elongated HDPE film consisting of a stacked structure, in which the fracture of crystalline lamellae is suppressed during stretching [48].

## 4. Conclusions

We found that strengthening occurred without sacrificing the strain recovery by the heat elongation of the TPU film consisting of a high HS content. The stress continuously increased with elongation ratio *λ*, i.e., the stress at break increased from 55 to 136 MPa using the heat elongation at *λ* = 300%, although the stress at break of the TPU film is usually below 50 MPa [11,14,23,26,41], and the literature on a high-strength TPU that can achieve a stress at break above 100 MPa is limited [29]. That is, a high-strength TPU elastomer could be obtained by heat elongation. The anisotropically shaped HS domains were stacked at the nanometer scale, and the longer direction of the HS domains was arranged perpendicular to the elongated direction by the heat elongation. The densification of the HS domain increased without a significant increase in the crystallinity with *λ*. Owing to the increase in the densification of the HS domains in the stacked structure, the increase in the stress at the stress upturn became larger with *λ*. Thus, the strengthening of the heat-elongated TPU might be attributed to the densification of the anisotropically shaped HS domains in the stacked structure, which prevents the fracture of the HS domains during the stretching. This strengthening concept of the stacked domain structure differs from the familiar strengthening concept of TPU, which requires fibrillation, strain-induced crystallization, and an increase in the crystallinity of the HS domains. This high-strength TPU is promising for applications such as industrial belt and medical tube applications, in which a light weight and thin form are desired.

## Figures and Tables

**Figure 1 polymers-14-01470-f001:**
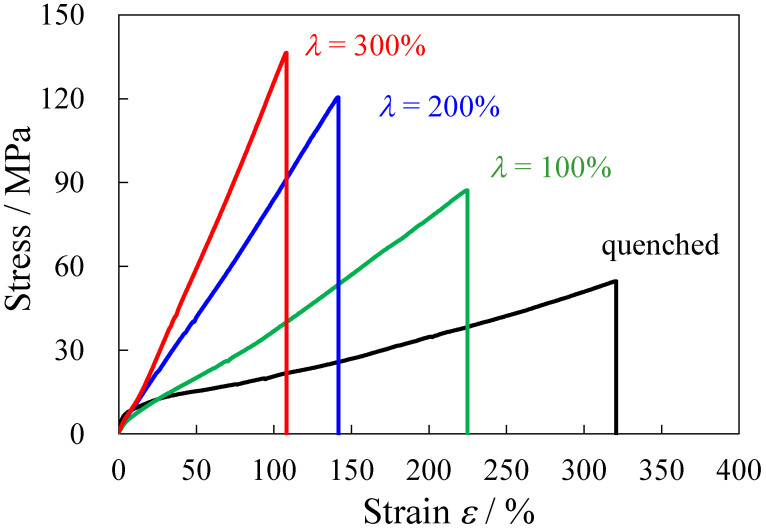
Stress–strain curves of the heat-elongated TPU films obtained at various elongation ratios *λ*.

**Figure 2 polymers-14-01470-f002:**
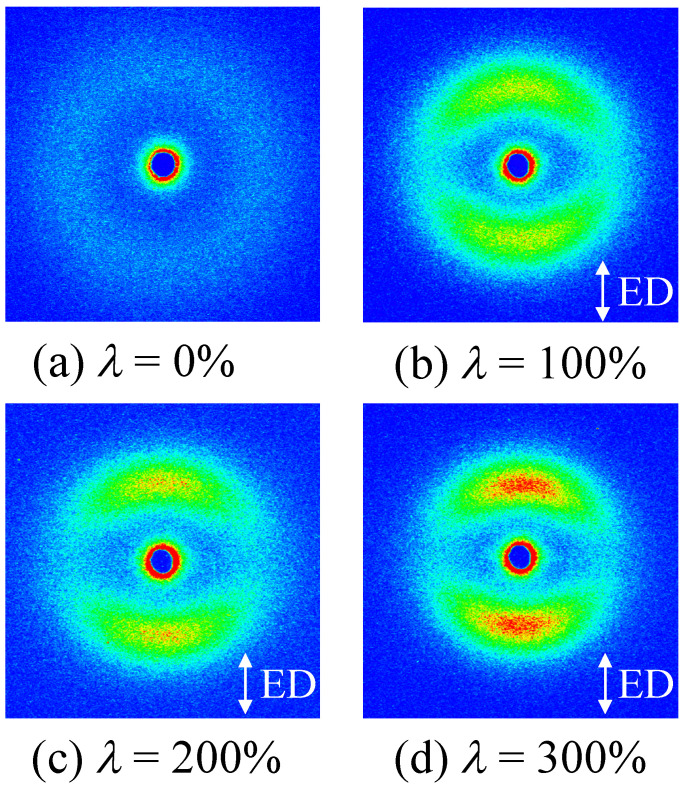
Two-dimensional SAXS patterns of the heat-elongated TPU films obtained at 150 °C at various elongation ratios *λ*; (**a**) *λ* = 0%, (**b**) *λ* = 100%, (**c**) *λ* = 200%, (**d**) *λ* = 300%.

**Figure 3 polymers-14-01470-f003:**
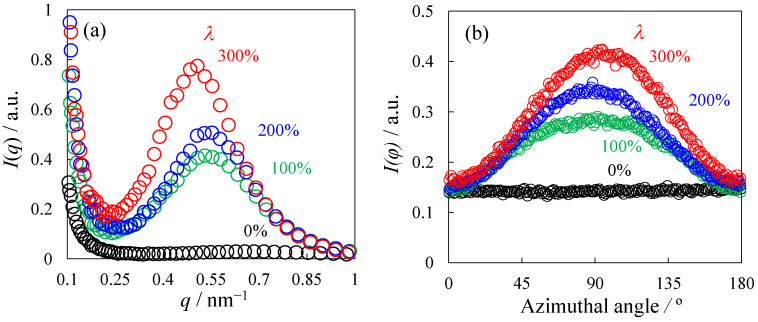
One-dimensional SAXS profiles of the heat-elongated TPU films obtained at various elongation ratios *λ*: (**a**) *q* dependence, (**b**) azimuthal angle dependence.

**Figure 4 polymers-14-01470-f004:**
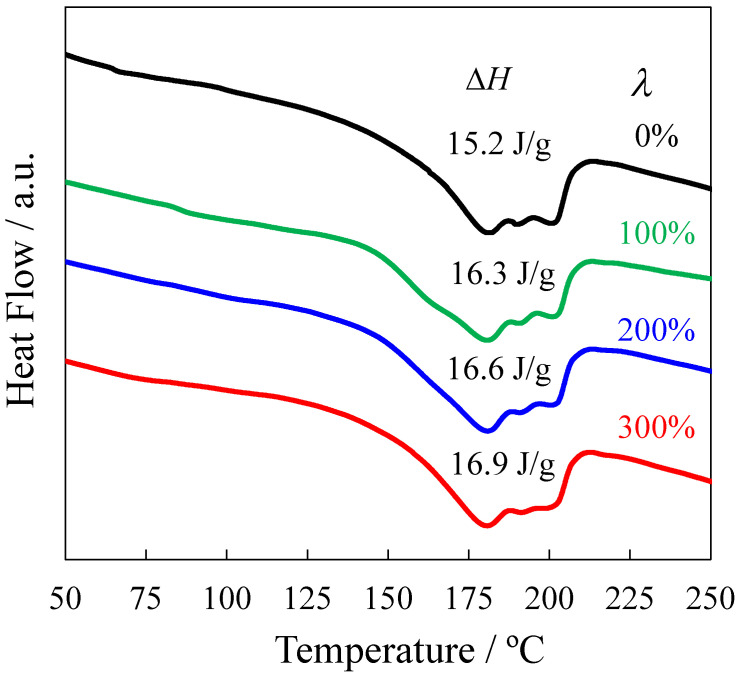
DSC curves for the melting peak of heat-elongated TPU films obtained at various elongation ratios *λ*.

**Figure 5 polymers-14-01470-f005:**
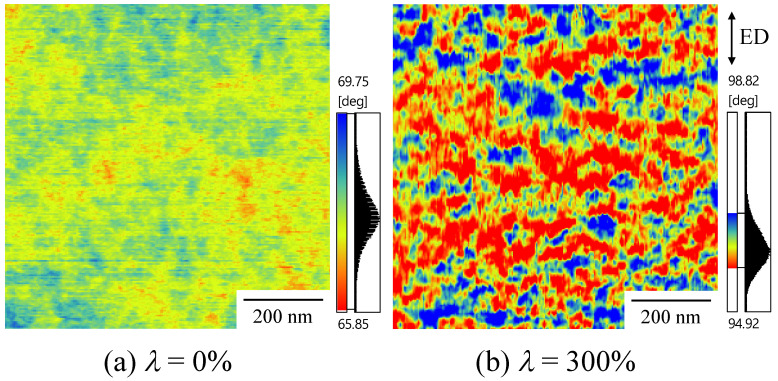
AFM phase images: (**a**) quenched TPU film (*λ* = 0%), (**b**) heat-elongated TPU film (*λ* = 300%).

**Figure 6 polymers-14-01470-f006:**
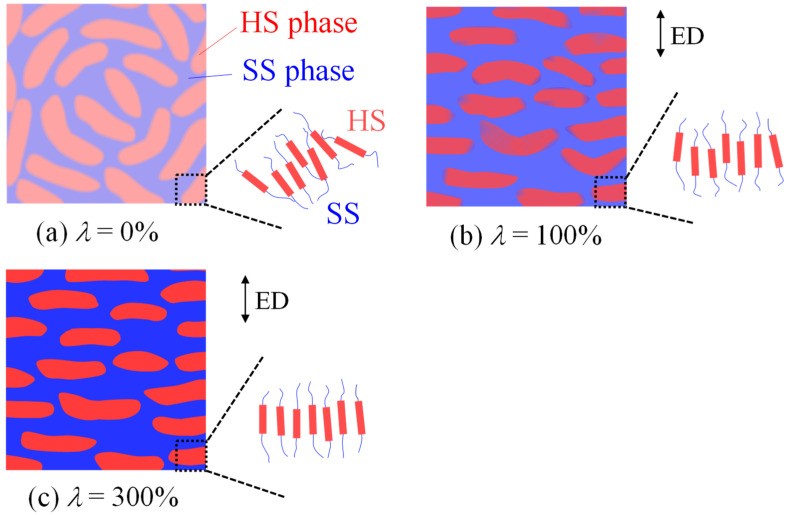
Schematic illustration of the heat-elongated TPU films obtained at 150 °C at various elongation ratios *λ*: (**a**) *λ* = 0%, (**b**) *λ* = 100%, and (**c**) *λ* = 300%. The contrast becomes stronger as the density difference increases.

**Figure 7 polymers-14-01470-f007:**
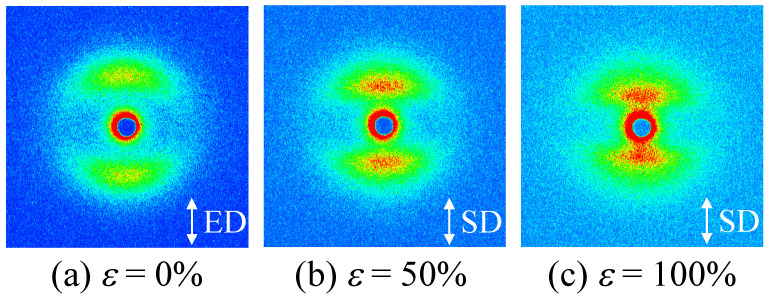
Two-dimensional SAXS patterns of the heat-elongated TPU film (*λ* = 300%) during uniaxial stretching at various strains ε at 23 °C; (**a**) *ε* = 0%, (**b**) *ε* = 50%, (**c**) *ε* = 100%. ED is elongated direction and SD is stretching direction.

**Figure 8 polymers-14-01470-f008:**
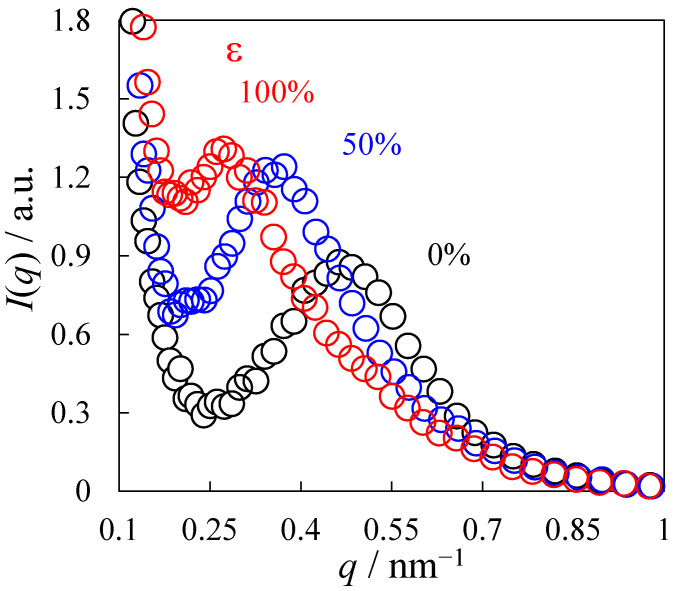
SAXS profiles of the heat-elongated TPU film (*λ* = 300%) during uniaxial stretching at various strains ε at 23 °C.

**Figure 9 polymers-14-01470-f009:**
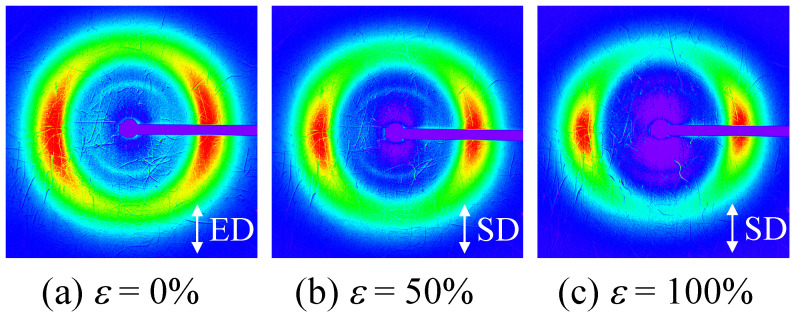
Two-dimensional WAXD patterns of the heat-elongated TPU film (*λ* = 300%) during uniaxial stretching at 23 °C at various strains ε; (**a**) *ε* = 0%, (**b**) *ε* = 50%, (**c**) *ε* = 100%.

**Figure 10 polymers-14-01470-f010:**
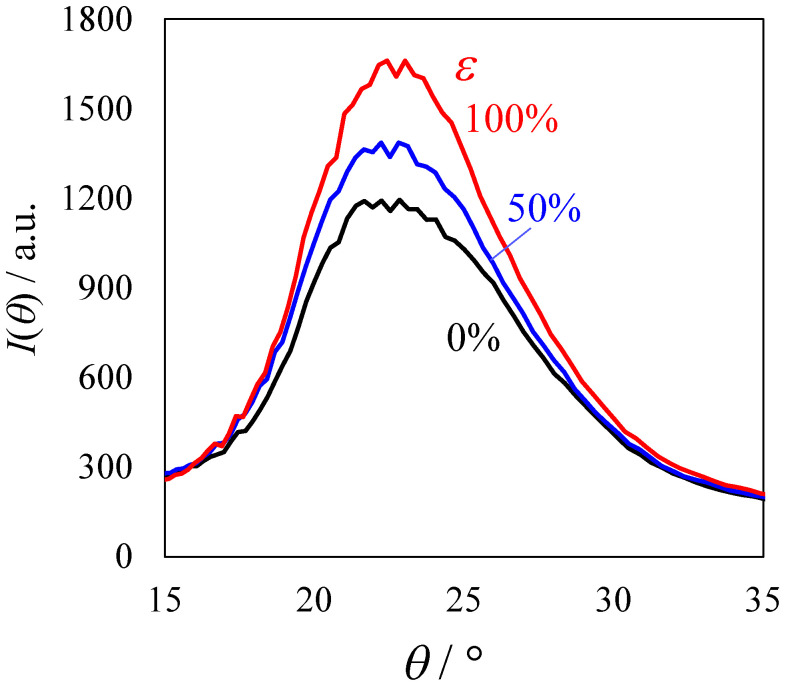
One-dimensional WAXD profiles of the heat-elongated TPU film (*λ* = 300%) during uniaxial stretching at 23 °C at various strains ε.

## Data Availability

Not applicable.

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
