# Peer review of "High-Strength Heat-Elongated Thermoplastic Polyurethane Elastomer Consisting of a Stacked Domain Structure"

_polymers, 2022, doi:10.3390/polym14071470_

Round 1
Reviewer 1 Report
This paper presented an investigation of a type of thermoplastic polyurethane elastomers and their related properties. The authors first briefly reviewed the state of the art. Then the materials, manufacturing process of thin films, and procedures to characterize materials properties were presented. The authors reported an adequate amount of results. However, some details of the reported work are missing in the current draft. The reviewer suggests publishing this paper after a revision.
- Could the authors clarify the novelty of this paper in the introduction? Why the reported work is different from other reported polyurethane work?
- Regarding the thin film tensile tests, did the dumbbell-shaped sample follow any standard? If so, please provide the standard number. concentration during testing?
- In the DSC test, was the sample heated in one cycle or two-cycles? Please provide more experimental details.
Reviewer 2 Report
Dear authors
I have overall enjoyed article Reading. The topic discussed by the authors is interesting to the audience, and in general, the article organization is OK. I list below some major and minor changes that must be replied one by one before article acceptance and publication:
Major:
The main findings of the proposed method have been detailed and explained along the manuscript with sufficient details. However, I do not see a possible application for this development. Could you please mention some possible applications for the high strength polyurethane? What would be the main motivation to manufacture this type of polyurethane?
The main contribution from this study lies on the specimen preparation. Nonetheless, some details are missing in section 2.1. They are next listed: line 77) what are the exact values for the “various elongation ratios”?, line 79) Were the TPU films naturally of force cooled?
Minor and English corrections
A broad review of the English grammar is required. I list below some corrections, but there are many more along the manuscript.
Lines 9 through 12: The sentence starting with “ The stress increase…” is just too long. Please consider splitting.
Line 113: The article “the” must be inserted after “was”, i.e. “specimen was the same to…”
Line 123 through 127: In these lines the word “increase” is employed a total of 7 times in adjective and verb forms. Please consider sentence rewriting.
Line 33: Although it may be obvious for the authors, please specify what kind of modulus is discussed in line 33.
Round 2
Reviewer 1 Report
The authors have revised the paper following the reviewer's comments. The reviewer suggests publishing this paper.
Reviewer 2 Report
Mandatory changes have been applied. The article can be published in present form